# Molecular Mechanisms in Pathophysiology of Mucopolysaccharidosis and Prospects for Innovative Therapy

**DOI:** 10.3390/ijms25021113

**Published:** 2024-01-17

**Authors:** Yasuhiko Ago, Estera Rintz, Krishna Sai Musini, Zhengyu Ma, Shunji Tomatsu

**Affiliations:** 1Nemours Children’s Health, 1600 Rockland Rd., Wilmington, DE 19803, USA; yasuhiko.ago@nemours.org (Y.A.); krishnasai.musini@nemours.org (K.S.M.); zhengyu.ma@nemours.org (Z.M.); 2Department of Molecular Biology, Faculty of Biology, University of Gdansk, 80-308 Gdansk, Poland; estera.rintz@ug.edu.pl; 3Department of Biological Sciences, University of Delaware, Newark, DE 19716, USA; 4Department of Pediatrics, Graduate School of Medicine, Gifu University, Gifu 501-1112, Japan; 5Department of Pediatrics, Thomas Jefferson University, Philadelphia, PA 19144, USA

**Keywords:** mucopolysaccharidosis, enzyme replacement therapy, gene therapy, innate immunity, immunomodulatory drugs

## Abstract

Mucopolysaccharidoses (MPSs) are a group of inborn errors of the metabolism caused by a deficiency in the lysosomal enzymes required to break down molecules called glycosaminoglycans (GAGs). These GAGs accumulate over time in various tissues and disrupt multiple biological systems, including catabolism of other substances, autophagy, and mitochondrial function. These pathological changes ultimately increase oxidative stress and activate innate immunity and inflammation. We have described the pathophysiology of MPS and activated inflammation in this paper, starting with accumulating the primary storage materials, GAGs. At the initial stage of GAG accumulation, affected tissues/cells are reversibly affected but progress irreversibly to: (1) disruption of substrate degradation with pathogenic changes in lysosomal function, (2) cellular dysfunction, secondary/tertiary accumulation (toxins such as GM2 or GM3 ganglioside, etc.), and inflammatory process, and (3) progressive tissue/organ damage and cell death (e.g., skeletal dysplasia, CNS impairment, etc.). For current and future treatment, several potential treatments for MPS that can penetrate the blood–brain barrier and bone have been proposed and/or are in clinical trials, including targeting peptides and molecular Trojan horses such as monoclonal antibodies attached to enzymes via receptor-mediated transport. Gene therapy trials with AAV, ex vivo LV, and Sleeping Beauty transposon system for MPS are proposed and/or underway as innovative therapeutic options. In addition, possible immunomodulatory reagents that can suppress MPS symptoms have been summarized in this review.

## 1. Introduction

Mucopolysaccharidoses (MPSs) are a group of inborn errors of the metabolism caused by a deficiency in the lysosomal enzymes required to break down molecules called glycosaminoglycans (GAGs), which are long, linear, negatively-charged polysaccharides composed of repeating disaccharide units [1]. Since the responsible enzyme resides in lysosomes, MPS is classified as a group of lysosomal storage diseases (LSDs). The incidence of MPS is estimated to be about 1 in 20,000 live births [2,3,4,5]. Considering that the incidence of LSDs is thought to be from 7.6 to 19.3 per 100,000 people [6,7], MPS makes up a significant percentage of LSDs. GAGs are present in various tissues comprising brain [8], visceral organs [9], bone, cartilage, tendons, eyes, skin, and other connective tissues [10]. Their molecular structure determines their functions in the body. The step-by-step breakdown of the terminal sulfate, acidic, and amino sugar residues by various lysosomal enzymes is necessary for the metabolic recycling of GAGs [11]. Patients with MPS do not produce one of the lysosomal enzymes necessary to break down these GAGs or generate enzymes that do not work appropriately [11], depending on the type of genetic mutations on each responsible gene, e.g., nonsense mutation, missense mutation, deletion, etc. These GAGs accumulate over time in multiple tissues, leading to progressive, irreversible cytotoxic damage that impacts cognitive function, growth, appearance, physical performance, and function of organs [12]. To date, eight distinct clinical types and subtypes of MPS III (A, B, C, D) and IV (A, B) have been identified, and 12 diseases have been classified as MPS. Most MPS types, except MPS IV and VI, have primary central nervous system (CNS) involvement, while most patients, except those with MPS III, have progressive systemic skeletal dysplasia [13]. Each subtype of MPS is associated with a deficiency of a particular enzyme, leading to the accumulation of specific types of GAGs, as described below. In 2014, Dr. Gurinova et al. [14] reported a novel disease of impaired GAG metabolism without a deficiency of known lysosomal enzymes: mucopolysaccharidosis-plus syndrome (MPSPS: OMIM #617303) [15,16]. MPSPS is an autosomal recessive multisystem disorder caused by a specific mutation, p.R498W, in the vacuolar protein sorting-associated protein 33A (VPS33A) gene. The name of the disease, MPSPS, means that in addition to typical symptoms for conventional MPS I, patients develop other features such as congenital heart defects and renal and hematopoietic disorders. It remains unknown how the missense mutation p.R498W in VPS33A causes the accumulation of GAGs. The detailed mechanisms and disease pathophysiology remain to be elucidated [17].

Knowledge of the disease stage at the start of treatment is essential in clinical trials before irreversible conditions occur in the CNS and bone. Hence, given the limitations of traditional therapies (intravenous ERT and HSCT; see below), the therapeutic approaches to the CNS and skeletal system still have unmet medical needs.

Fratantoni et al. observed the phenomenon of complementary correction in vitro by culturing both fibroblasts from patients with Hunter syndrome (MPS II) and those with Hurler syndrome (MPS I) together [18], which led them to postulate that there are molecules passed from one cell to another, serving to ameliorate the pathological phenotype. Two years later, this unknown molecule was assumed to be a protein based on the results of polyacrylamide gel electrophoresis [19], and further purification and characterization followed [20]. Finally, in 1973, using a radioactive mucopolysaccharide, it was proven that a deficiency in sulfoiduronate sulfatase was the cause of Hunter syndrome [21]. These ideas and findings are the basis of hematopoietic stem cell transplantation (HSCT) and enzyme replacement therapy (ERT) for inherited metabolic diseases, leading to the first successful HSCT in the field of MPS in 1981 [22,23]. However, in HSCT, several drawbacks are present: difficulties in finding an HLA-matched suitable donor, severe complications related to the suppression of bone marrow, and/or graft-versus-host disease [23].

In 2003, α-L-iduronidase (Aldurazyme) with weekly intravenous injections was approved to treat MPS I [24], followed by other ERTs for MPS VI [25], II [26], IVA [27], and VII [28], although these traditional ERTs could not provide a solution for CNS impairment and skeletal dysplasia. In addition, patients need to receive weekly or bi-weekly infusions, and the cost of ERT cannot be ignored.

The selectively permeable blood–brain barrier (BBB) is a significant hurdle [29], but understanding the unique initial neuro-pathophysiology for MPS is critical to success. The earlier treatment is initiated, the better the outcome for the CNS in MPS [30]. Early intervention is based on accurate prognosis and validated biomarkers, although these are beyond the scope of the manuscript. Thus, treating the CNS is an area of research currently receiving much attention. Although unsuccessful, the first clinical trials to approach CNS were via intrathecal (IT) ERT for MPS II and IIIA [31]. Successively, intracerebroventricular (ICV) ERT has been conducted and approved for MPS II, and phase 2 clinical trials are underway for MPS IIIB (ClinicalTrials.gov identifier: NCT03784287). Setting devices for IT/ICV administration and direct infusions into cerebrospinal fluid provide potential patient risks, including surgical procedures and infections. HSCT and IT/ICV ERT must be balanced with disease risks. Intravenous ERTs with molecular Trojan horses (monoclonal antibodies attached to enzymes) have been developed to penetrate the BBB with a low risk of adverse effects. This novel ERT is conducted via receptor-mediated transport [32,33], leading to its approval for MPS II [34] and successive clinical and preclinical trials for MPS I (ClinicalTrials.gov identifier: NCT04227600, NCT04453085) [35], IIIA [36], and IIIB [37].

The current approaches to treating MPS skeletal dysplasia are also insufficient. Many patients with MPS have experienced various orthopedic surgical interventions for skeletal problems; however, these do not remove the primary cause of MPS. Little attention has been focused on the direct and indirect pathophysiological consequences of involved cell types (osteoblasts, osteoclasts, osteocytes, chondrocytes) beyond those where storage materials preferentially accumulate [38]. Once threshold levels of the disease are exceeded in bone, its recovery is impossible, especially in the growth plate. 

To overcome these hurdles, the clinical applications of gene therapy with AAV, ex vivo LV, and gene editing for MPS have recently been considered, and several clinical trials are currently underway for MPS I, II, IIIA, IIIB, and VI [39]. This review will provide an updated insight into the pathogenesis of innovative therapies for MPS, focusing on the central nervous and skeletal systems.

Moreover, recent studies have revealed the profound relationship between immunity and MPS [40,41]. The involved inflammatory pathways are complex, and many molecules are closely related. In other words, there are still many possible targets to suppress to alleviate the symptoms of MPS patients. We have elucidated these complex signaling pathways and then pointed out candidate reagents to dampen the activated inflammation in this paper.

## 2. Accumulation of GAGs and Subsequent Pathophysiological Alterations

Symptoms of MPS are essentially caused by the accumulation of GAGs, resulting in multi-system dysfunction, including the CNS and skeletal system. GAGs consist of five types of disaccharides: heparan sulfate (HS), dermatan sulfate (DS), chondroitin sulfate (CS), keratan sulfate (KS), and hyaluronan (HA) [42]. Apart from HA, all GAGs attach to the core protein, forming proteoglycans [43]. Abnormalities in the brain and connective tissues, such as bones, joints, and ligaments, result from the storage of several substances in the cells that make up the organ or tissue [44]. These issues induce neurological symptoms, skeletal deformity, impaired movement of joints, and growth impairment [12,44].

Depending on the impaired enzyme type, the accumulated GAGs and degrees differ, leading to a specific disorder and diverse clinical manifestations (Table 1). The clinical severity of the disease is mainly determined by residual enzyme activity and the distribution and turnover of the substrates impacted by the deficiency [45]. Mild-to-intermediate forms are also known as “attenuated phenotypes”. The phenotype of affected siblings can differ due to unknown genetic factors [46]. Except for MPS II, which is X-linked and therefore has a relatively high prevalence compared to other types of MPS, especially in males, all MPSs are autosomal recessive diseases [47]. Typical symptoms of MPSs and their pathogenesis caused by each accumulating substance are described below.

### 2.1. Heparan Sulfate (HS)

HS, consisting of glucuronic acid and N-acetylglucosamine, is a ubiquitous compound present in both vertebrates and invertebrates [52]. HS can interact with various proteins, cytokines, and growth factors, depending on the sulfation of hydroxyl groups or the amino groups present in HS [53]. HS is linked to a protein core via a serine residue attached to a tetrasaccharide linker to form heparan sulfate proteoglycan (HSPG) [54], which can be classified into three groups based on their location: membrane, secreted to the extracellular matrix, and secretory vesicles in the cytoplasm. They play an essential role as a coreceptor in interacting with diverse protein ligands, including cytokines, to regulate various biological processes, such as immune reaction, angiogenesis, blood clotting, tumor metastasis, and developmental processes [55,56,57,58,59,60].

HS accumulation is responsible for multiple organ dysfunction in several MPS types (Table 1). The impairment of HS degradation has broad clinical manifestations, including cognitive impairment, hepatomegaly, and coarse faces. Clinical phenotypes may differ depending on the defective enzyme, but the main neurological symptoms are similar, such as cognitive retardation, aggressive behavior, hyperactivity, seizures, progressive dementia, vision loss, and sleep disorder. HS storage interferes with many processes, such as activation of the ligands and receptors, mitochondrial function, intracellular trafficking, or autophagy, which can result in neuronal death [61,62,63,64,65,66]. Subsequent accumulation of undegraded HS in the extracellular matrix or cellular membrane triggers several pathological mechanisms. For example, an MPS IIIB mouse model study showed that these HS fragments activated the TLR4 (Toll-like receptor 4)-Myd88 (myeloid differentiation primary response protein 88) pathway in microglia, which turns on innate immunity and induces neuroinflammation in the brain [67], which is discussed in more detail in a later section. Similarly, those fragments triggered activation of the focal adhesion process in astrocytes and neuronal stem cells in the MPS IIIB mouse model, modulating the integrin signaling pathway and axonal development, which may lead to disturbing neuronal cell behavior [66]. Also, HS accumulation in the brain may affect adult neurogenesis [68]. Additionally, increased sulfation in HS in the brain of the MPS mouse models decreased their capacity to interact with proteins or growth factors [69]. Abnormal sulfation of HS affects the interaction between HS and chemokines, which induces impaired migration of hematopoietic stem and progenitor cells after HSCT, as shown in the MPS I mouse model [70]. Moreover, abnormal HS in MPS I cannot act as a coreceptor for fibroblast growth factor (FGF) signaling, affecting neuronal signaling and function [71]. This can negatively affect neuronal development and neuroplasticity [72], resulting in glial cell death and neurodegeneration [73]. Deregulated FGF signaling is also associated with the development of skeletal defects, as confirmed in zebrafish and mouse models of MPS II [74].

Primary accumulation of HS in lysosomes can inhibit other enzymes in the lysosome, resulting in secondary storage materials. Using fibroblasts with MPS III A, B, C, and D patients, Lamanna et al. found an increase in intracellular dermatan sulfate (DS), not just HS. They also confirmed through in vitro studies that HS inhibits iduronate 2-sulfatase, the enzyme deficient in MPS II [75]. This finding was consistent with previous results from a diagnostic approach, where unexplained elevations of DS were detected in MPS III patient serum and urine [76]. Thus, it is well known that the primary accumulation of a specific substance in lysosomes can impair lysosomal function and interfere with other acid hydrolases, leading to the secondary accumulation of other substances that should be degraded in the lysosome [77]. In the case of MPS with CNS symptoms, the problematic secondary or tertiary accumulating substances would be gangliosides (GM2 and GM3) [78,79], globotriaosylsphingosine [80], unesterified cholesterol, which is a crucial structural element of the plasma membrane [81,82], tau protein, which is one of the autophagic substrates [83], and β-amyloid [41,84,85]. GM2, globotriaosylsphingosine, and unesterified cholesterol are also primarily accumulated in GM2 gangliosidosis, Fabry disease, and Niemann–Pick Type C disease, respectively [86,87,88], which implies that the secondary accumulation of these substances in MPS might also further exacerbate the neurological symptoms. Tau protein is primarily found in neurons and is involved in stabilizing microtubules, which are important for maintaining the structure of neurons [89,90]. Like other cellular proteins, tau protein is constantly produced and degraded in cells. The degradation of tau protein involves autophagy [91], by which cells break down and recycle their components, including proteins and organelles. Therefore, tau protein accumulation in MPS suggests impaired autophagy [92], which can also negatively impact neurological symptoms, as shown in several other diseases, such as Alzheimer’s, Parkinson’s, and Huntington’s diseases [93,94,95]. β-amyloid accumulation would also have a negative effect on the brain, as shown in previous studies [96].

HS also participates in bone morphogenesis, as shown by its interaction with bone morphogenetic proteins (BMPs) signaling pathways in MPS models [97,98,99]. BMPs participate in many processes during the development and homeostasis of organs like bone, blood vessels, cartilage, and muscles [99]. Overexpressed syndecan-3, one of the HSPGs on the cell membrane, inhibits the interaction of BMP2 with its receptors on mesenchymal cells, preventing chondrogenesis in vitro [100]. In contrast, exogenous HS enhances the ability of BMP2 to facilitate chondrogenesis [100]. When HS catabolism is inhibited in lysosomes, it would be likely that these two opposite effects could be facilitated and offset each other, making the effect of HS accumulation on skeletal symptoms small (Table 1).

### 2.2. Chondroitin Sulfate (CS) and Keratan Sulfate (KS)

CS consists of linear polymers of disaccharide units containing glucuronic acid and N-acetylgalactosamine. They are attached to a protein core of a proteoglycan by a serine residue and a tetrasaccharide linker, as is HS [101]. CS polysaccharide chains attached to proteoglycans range from 10 to 200 repeating disaccharides found on cell surfaces and in the extracellular matrix [102]. Depending on their sulfation, there are two main CSs: chondroitin-6-sulfate (C6S) and chondroitin-4-sulfate (C4S). Using densitometry, Mourão et al. measured these two disaccharides in human cartilage [103]. A total of 98% of the CS in adult articular cartilage (the proximal femoral epiphysis and the proximal tibial epiphysis) is C6S, and C6S is also dominant in vertebral and articular cartilage (the proximal and distal femoral epiphyses and the proximal tibial epiphysis) of children up to 2 years of age, whereas C6S and C4S are equally distributed in growth plate cartilage (the proximal and distal femoral growth plates and the proximal tibial growth plate) of children up to 3 years of age [104,105]. C4S is dominant in human dentin and bovine bone [106], and immunoelectron microscopic analysis showed that as calcification progresses, the rate of C4S increases in rat growth plate cartilage [107].

In summary, C6S is abundant in cartilage, while C4S is also found in cartilage and is important for bone and calcification. KS comprises galactose and N-acetylglucosamine, forming three polysaccharides, KS I, II, and III [108]. KS I is N-linked via asparagine in the core protein with a linkage containing three mannoses [108], mainly found in corneas [109]. KS II is abundant in cartilage and is bound to a serine or threonine residue using N-acetylgalactosamine as a linker, while KS III is found predominantly in brain and is attached to a serine or threonine residue using mannoses as a linker [54,108].

The extracellular matrix of connective tissue, such as cartilage, comprises collagen and a proteoglycan called aggrecan [110,111]. Collagen gives connective tissue tensile strength, whereas aggrecan helps absorb impact forces and lubricates joints [111]. Both CS and KS are the main GAGs that are components of aggrecan in cartilage [111,112]. Patients with MPS IVA cannot catabolize these two molecules, and MPS IVB patients cannot degrade KS. This is why cartilage lesions are a characteristic manifestation of MPS IV. The sulfate and carboxylate groups of these two GAGs are negatively charged and attract water molecules to the aggrecan, and then these pulled water molecules work together with the aggrecan to absorb physical impacts [111]. Impaired KS and/or C6S degradation damages cartilage, including growth plates [113], making disturbed bone growth in MPS IV patients refractory to enzyme replacement therapy, as discussed in a later section. In contrast, what accumulates in patients with MPS VI is C4S, which is abundant in both growth plates and bones, as described in the previous paragraph. In addition, another accumulating GAG, dermatan sulfate (DS), causes additional abnormality in bone, as described in the following section, leading to skeletal dysplasia in MPS I, II, VI, and VII. 

In addition to acting as a cushion in cartilage, CS has other properties; for example, it can stimulate the production of proteoglycans and type II collagen in articular cartilage [114] and attenuate nuclear factor-κB (NF-κB) signaling, thereby reducing inflammation in the synovial membrane and chondrocytes [115]. Accumulation or abnormal sulfation of CS may also affect these aspects. Type IIA collagen N-propeptide (PIIANP) was decreased in the serum of MPS IVA patients [116]. In contrast, a low expression of collagen type II and a high expression of collagen type I were observed in the distal femoral articular cartilage of two adult MPS IVA patients (35 and 38 years old) using immunohistochemistry and real-time PCR [117]. Moreover, Bank et al. reported changes in molecular morphology within the extracellular matrix of articular cartilage from the knee joints of two adult female patients with MPS IVA (42 and 31 years old) using electron microscopy [118]. On the other hand, cytokines induced by the NF-κB pathway, namely IL-1β and IL-6, are elevated in the serum or plasma of untreated patients with MPS IVA [116]. These pro-inflammatory factors in the bloodstream may exacerbate systemic inflammation, including arthritis in MPS.

Regarding KS, its accumulation is associated with cartilage lesions and ophthalmologic symptoms [117,119]. The regulation of collagen fibril spacing, which is crucial for optical clarity, is one of the functions of KS in the cornea. Mutations in the *GALNS* or *GLB1* genes can cause abnormal KS sulfation patterns, which may lead to corneal opacity and impaired vision in MPS IV patients [108].

### 2.3. Dermatan Sulfate (DS)

DS consists of iduronic acid (IdoA) and N-acetylgalactosamine (GalNAc). Like HS and CS, they are attached to a protein core of a proteoglycan by a serine residue and a tetrasaccharide linker [101]. This carbohydrate is found primarily in the skin, heart valves, blood vessels [120], and bones [121]. Even though HS, CS, and KS are also found to be normal components of heart valves and large vessels [122,123,124], DS is a predominant component of normal heart valves [123]. Thus, there is a strong correlation between the accumulation of DS (MPS I, II, and VI) and pathological changes in the cardiac valves [125,126] (Table 1).

In bones, DS constitutes several proteoglycans together with CS, for example, decorin and biglycan [121], which are a member of the small leucine-rich proteoglycan family and are composed of a core protein with leucine repeats and a chain of CS and DS glycosaminoglycans that are covalently attached [127]. DS side chains of decorin wrap just outside each collagen fiber, forming a ring-mesh-like structure to bundle collagen fibers pointing in the same direction in the extracellular matrix [127,128,129], while biglycan appears to be involved in forming apatite in bone [130]. Both proteoglycans are expressed during bone formation, and once mineralization begins, only CS constitutes side chains of these proteoglycans, as observed in in vitro experiments using a rat model [131]. After bone formation, biglycan with CS side chains holds water molecules in the bone and maintains bone stiffness [132]. Considering these observations and noting that skeletal dysplasia is more prominent in MPS I, II, VI, and VII than in MPS III, there would be a strong correlation between DS accumulation and skeletal dysplasia.

DS is also present in the cornea [120], where symptoms appear when DS accumulates, as seen in MPS I, VI, and VII. However, in the case of MPS II, corneal opacity does not occur [133]. The deficiency in the IDS enzyme in MPS II results in the build-up of DS containing an additional sulfate group (C-4 and C-2) as compared to MPS I and VI (C-4 only). Tomatsu et al. hypothesize that the additional sulfate group on the DS in MPS II exhibits a protective effect in preventing corneal clouding [134].

A recently discovered novel type is MPS X [49]. This is caused by a deficiency in arylsulfatase K (Arsk), which works as a 2-sulfoglucuronate sulfatase to degrade HS, CS, and DS [135,136]. In a mouse model, Arsk deficiency caused a significant accumulation of HS [136]. Nonetheless, four out of the six patients reported were tested for urinary GAG; all showed elevated DS, and two of them showed opacity of the lens and vitreous bodies, while only one showed a slight elevation of HS [49,137]. The cause of this discrepancy between humans and mice is not well elucidated; however, human cases of MPS X also imply that DS is strongly associated with ophthalmologic symptoms.

### 2.4. Hyaluronic Acid (HA)

Glucuronic acid and N-acetylglucosamine form the disaccharide unit of HA. A membrane-bound enzyme, hyaluronan synthase, alternatively links uridine diphosphate N-acetylglucosamine and uridine diphosphate glucuronic acid in the cytoplasm; then, the synthesized long polysaccharide chain, HA, is ejected into the extracellular matrix [138]. No sulfation is necessary for HA to work in living things. In humans, HA is primarily present in the vitreous body and synovial fluid and is also found in connective, epithelial, and neural tissues to hold water molecules in these organs [139,140]. HA is broken down by hyaluronidase [141], and several enzymes have been revealed to have hyaluronidase activity. However, a deficiency in hyaluronidase 1, encoded by the *HYAL1* gene, is thought to cause MPS IX [142]. Only four cases of MPS IX in humans have been reported [50,142,143]; thus, it remains a challenge to determine a detailed picture of this disease. However, as far as we know, arthritis seems to be the main symptom of MPS IX, which was also the case in the mouse model of this disease [144]. Given its characteristic ability to attract water molecules [145], it is not surprising that the accumulation of HA in synovial fluid can swell joints.

### 2.5. Symptoms Associated with Several Types of GAGs

Dysostosis multiplex refers to a set of distinctive bone anomalies [146]. Due to insufficient remodeling, the long bones are short and can be thick with an irregular hyperostotic shaft and metaphysis in MPS I, VI, and VII. The distal epiphyses of the radius and ulna are tilted and angled abnormally. The clavicle has widened ends. Flared iliac bones, a flattened acetabulum, and a coxa valga deformity are present. However, if we observe closely, there is variability in the “dysostosis multiplex” depending on the different types of GAGs that accumulate. The accumulation of DS and C4S has direct detrimental effects on bone, as we have already discussed in the previous sections. Although the adverse effect of the buildup of C6S and KS is primarily on the cartilage, damage to the growth plates would undoubtedly harm the growth and volume of bone. This difference in molecular pathophysiology is what makes MPS IV symptoms unique. Most MPS patients with skeletal symptoms tend to have thick bones, mainly due to DS and/or C4S accumulation in the bones, resulting in stiff joints. On the other hand, patients with MPS IV show a laxity of the joints, mainly due to the smaller volume of each bone in MPS IV compared to other types of MPS [147]. Considering that C4S and DS do not primarily accumulate in MPS IV, the difference in bone volume is understandable. This is also true in the case of vertebrae since a vertebra grows with two growth plates like other long bones. MPS IVA patients tend to have smaller vertebrae than those with other types of MPS with prominent enlarged interpediculate spaces, showing universal platyspondyly with central anterior beaking of each vertebra. This feature contrasts with MPS I and II, which usually show anteroinferior beaking and posterior scalloping of the lumbar vertebrae [148]. Dysostosis multiplex is found in MPS I, II, IV, VI, and VII, but it is less common in MPS III and attenuated variants [149], which suggests that the accumulation of GAGs other than HS is more strongly involved in this symptom.

## 3. Activated Inflammatory Pathways in MPS: Potential Molecular Targets for Therapeutic Intervention (Figure 1)

Recent studies have provided insight into the link between storage materials, lysosomal dysfunction, innate immune activation, and subsequent excessive inflammation exacerbating MPS symptoms. As these mechanisms would be important targets for novel therapies, these relationships are summarized in this section.

In lysosomal storage diseases (LSDs), the excessive accumulation of substances in lysosomes subsequently leads to lysosomal disruption, where the integrity of the lysosomal membrane is compromised [150,151,152], causing the leakage of lysosomal enzymes, including cathepsin B, which can augment the inflammasome [153,154]. In addition, lysosomal dysfunction prevents the normal degradation process and contributes to impaired autophagy, which requires functional lysosomes and is the cell’s ability to break down and remove dysfunctional cellular components [155]. Flawed autophagy can also contribute to increased IL-1β secretion via the decreased degradation of pro-IL-1β [156].

The impaired autophagy affects general cellular components and specialized organelles like mitochondria, leading to impaired mitophagy [157,158]. Mitophagy is the selective degradation of mitochondria, and when this process is compromised, mitochondrial dysfunction occurs [159,160,161]. This dysfunction further exacerbates cellular stress and increases reactive oxygen species (ROS) [160], highly reactive molecules that can damage cellular components. Moreover, because lysosomal activities such as acidification by the proton pump V-ATPase depend on mitochondrial ATP production, mitochondrial dysfunction can further worsen lysosomal function [162].

An increased level of ROS triggers an increase in thioredoxin-interacting protein (TXNIP) dissociated from thioredoxin (TRX) [163,164]. TRX normally keeps TXNIP inactive by binding to it. However, under oxidative stress, TXNIP is dissociated from TRX and becomes active. This dissociation plays a role in activating NOD-, LRR-, and pyrin-domain-containing protein 3 (NLRP3) [163], a protein that forms a part of the inflammasome complex responsible for initiating inflammatory responses. The activated NLRP3 inflammasome, in turn, leads to the production and release of pro-inflammatory cytokines, inducing inflammation at the cellular and systemic levels [165].

Another critical trigger for amplified inflammation in MPS is the TLR4 signaling pathway, originally for the innate immune response against lipopolysaccharide (LPS) generated by Gram-negative bacterial infections. This signaling pathway in microglia or peripheral blood mononuclear cells, which involves the activation of the MyD88 adaptor protein, causes inflammatory processes in the brains of MPS patients, mainly triggered by HS accumulation, leading to the expression of proinflammatory cytokines [67,166] (Figure 1). Oxidative stress induced by lysosomal dysfunction also enhances this pathway via the activation of NLRP3, as described in the previous paragraph (Figure 1). Many aspects of HS interactions with TLR4 remain unclear; however, HS stimulates this receptor for innate immunity from the extracellular space (Figure 1) because of the structural similarities between HS fragments and bacterial LPS, the original ligand of TLR4 [167]. The sulfation level of HS also matters. As the degree of sulfation in HS increases, the stimulation to TLR4 is stronger, judging from the neuropathology in MPS I, IIIA, and IIIB mouse models [68,69]. In an MPS I murine model, N-deacetylase/N-sulfotransferase (NDST), which adds a sulfate group to the amino group of glucosamine within the heparan sulfate chain, was more active than normal, increasing stimulation to TLR4 [64]. In the case of HA accumulation in MPS IX, its involvement in the TLR4 signaling pathway still seems controversial, as the length of the saccharide chain may also matter, and several studies have demonstrated inconsistent results [168,169,170,171,172]. Gangliosides, which were shown to accumulate in MPS III animal models [173,174], can also trigger the TLR4 receptor as a secondary storage substance [175,176]. Clinical evidence from the cerebrospinal fluid of MPS I patients is consistent with these theories, showing elevated levels of cytokines downstream of TLR4 [177]. Not only in CNS, the TLR4 signaling pathway may also be the primary cause of activated inflammation in the cardiovascular system of MPS I and VII patients [178,179] as well as in the osteoarticular system of patients with MPS I, II, VI, and VII [180]. Therefore, this signaling pathway initiated by TLR4 would have a critical role in MPS. However, it is not the whole mechanism because even when TLR4 was knocked out in an MPS IIIB mouse model, the transcription of IL-1β mRNAs in microglial cells was not stopped at 8 months [67]. Also, regarding bone growth, two MPV VII mouse models with TLR4 knocked out showed conflicting results [181,182]. More investigation is needed to elucidate the pathways associated with inflammation in MPS.

The role of CS in inflammation in MPS is debatable. CS is already approved as a therapeutic agent for the treatment of osteoarthritis in some countries, and several studies have reported its anti-inflammatory effect [183,184,185]. Tan and Tabata demonstrated the anti-inflammatory effects of exogenous C6S at concentrations of 50–100 μg/mL and C4S at 50 μg/mL in vitro using mouse macrophages [186]. Interestingly, this phenomenon was not completely dose-dependent, since at concentrations of 200 μg/mL or higher, C6S did not show such an effect or showed raised NO synthesis as a marker of innate immune response [186]. On the other hand, subsequent reports of human cases observed increased oxidative stress and pro-inflammatory factors in the blood of MPS VIA patients [116,187,188]. Taken together, it could be true that administered extrinsic CS often exerts an anti-inflammatory effect. However, in the case of intrinsic CS that has already filled the lysosomes of MPS patients, the pro-inflammatory effect caused by lysosomal dysfunction may offset and even surpass the anti-inflammatory effect of CS.

Activation of the innate immune response ultimately leads to the secretion of IL-1β, IL-6, TNF-α, IL-18, or matrix metalloproteinases (MMPs) by immune cells, including microglia, monocytes, and macrophage, with the activated NF-κB signaling pathway and NLRP3 inflammasome [189,190,191,192,193,194,195,196,197,198,199,200,201,202,203,204,205,206]. MMPs degrade the extracellular matrix of various tissues, including the skeletal system [207], which may be related to the symptoms of MPS [208]. Once these secreted cytokines bind to their respective receptors, activated cells (for example, chondrocytes, synovial cells, periodontal ligament fibroblasts, cardiomyocytes, and squamous epithelial cells) can also express MMPs [209,210,211,212,213,214,215,216]. Buffolo et al. reported that IL-1β reduced the excitatory synapses of mouse neurons in vitro with decreased frequency and amplitude of spontaneous synaptic currents, decreased density of excitatory synaptic connections, and decreased frequency of action potential-evoked Ca2+ transients [217]. IL-1β and TNF-α can induce the compromise and apoptosis of astrocytes [218]. Moreover, secreted IL-18 may induce positive feedback of the inflammatory pathway via MyD88 and NF-κB because macrophages and microglia express IL-18 receptors on their cell membrane [195,219,220,221,222,223,224,225] (Figure 1).

In summary, like other LSDs, lysosomal dysfunction in MPS leads to elevated cytokine levels via multiple pathways, resulting in activated systemic inflammation (Figure 1). HS and gangliosides accumulated in the extracellular matrix can amplify this inflammation via TLR4. Due to the complexity of these inflammatory pathways, there could be a plethora of targets to suppress inflammation, as described in later sections. As mentioned so far, since the activated inflammation can affect almost the entire body, anti-inflammatory treatments would be necessary for both skeletal and neurological symptoms.

**Figure 1 ijms-25-01113-f001:**
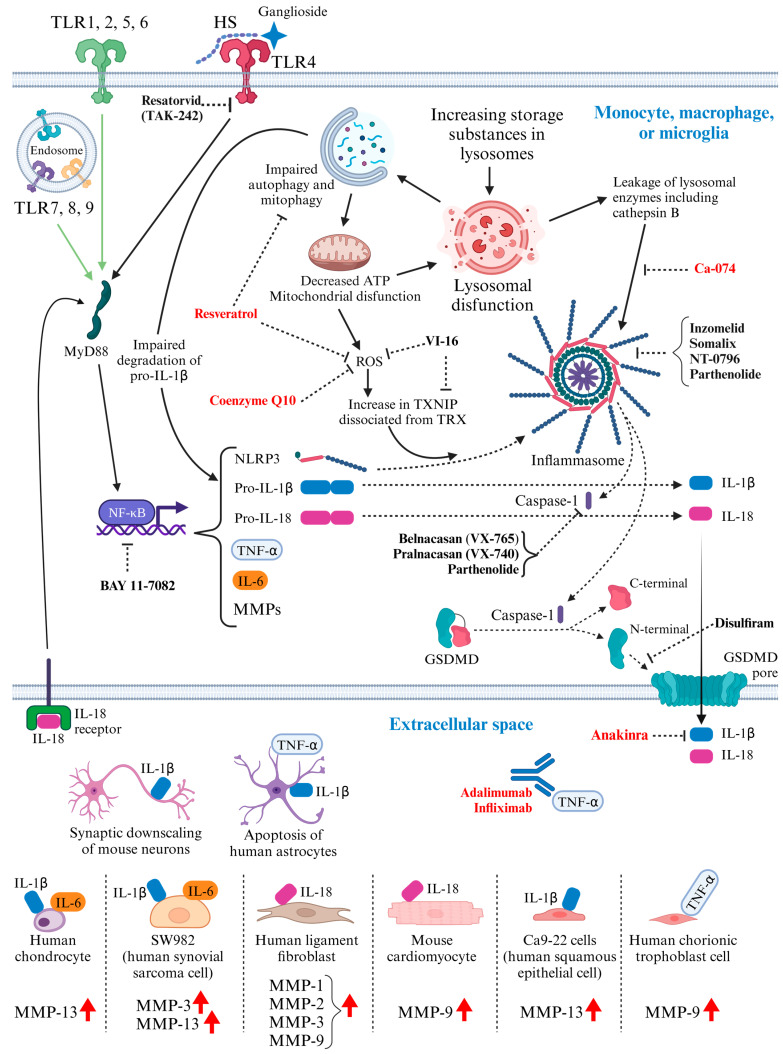
Representative inflammatory pathways and activated innate immunity in MPS [226,227]. Black arrows indicate activated pathways in MPS. Green arrows indicate pathways that are not considered to be activated. Red arrows indicate enhanced expression of each substance. Short, flat vertical lines at the tips of the dotted lines indicate inhibition by the reagent. Reagents that have been tested in MPS models are colored red. HS: heparan sulfate; TLR: Toll-like receptor; MyD88: myeloid differentiation primary response protein 88; NF-κB: nuclear factor kappa-light-chain-enhancer of activated B cells; ATP: adenosine triphosphate; ROS: reactive oxygen species; TXNIP: thioredoxin-interacting protein; TRX: thioredoxin; NLRP3: NOD-like receptor family pyrin domain-containing 3; IL: interleukin; TNF-α: tumor necrosis factor-alpha; GSDMD: gasdermin D; MMPs: matrix metalloproteinases. This figure was created with Biorender.com (accessed on 1 December 2023).

## 4. Development of Innovative Therapies

### 4.1. Hurdles for Effective ERT in CNS and Bone

Delivering the enzyme to target tissues is essential for successful ERT. To treat neurological symptoms, enzymes administered intravenously need to cross the BBB to reach the central nervous system, which is especially important for MPS I, II, III, and VII. HSCT also provides the enzymes from the transplanted stem cells. Additionally, to some extent, HSCT can be regarded as one of the strategies to overcome the BBB because hematopoietic stem cells can cross the BBB and differentiate into microglia to produce the deficient enzyme within the CNS [228,229,230,231,232]. Several long-term clinical reports support this perspective [233,234,235]. However, there are practical difficulties in finding a suitable donor for HSCT. Moreover, HSCT has not been effective in past cases of MPS III [236,237]; therefore, a more robust strategy to overcome the BBB is required.

Receptor-mediated transcytosis (RMT) [238], often depicted as a “Molecular Trojan Horse” [239], is currently the most promising way to overcome the BBB from a molecular biological standpoint, and interest in RMT is not limited to the field of LSD [240]. Endothelial cells of capillaries in brain have receptors to transport essential molecules from the blood stream to the central nervous system. Among such receptors, the insulin receptor was targeted in preclinical studies for MPS IIIA and IIIB [36,37] and clinical trials for MPS I and II (ClinicalTrials.gov identifier: NCT03053089 and NCT02262338, respectively). However, in the former clinical trial, 6.4% of MPS I patients experienced drug-related transient hypoglycemia [241], which caused concern.

Then, another focus of attention is the transferrin receptor (TfR). The original role of TfRs on the endothelial cells of capillaries in the human brain is to mediate the uptake of transferrin-bound iron from the bloodstream into the brain tissue. Iron is a critical nutrient for the brain and is required for various processes such as neurotransmitter synthesis, myelin synthesis, and energy metabolism [242] (Figure 2). By binding molecules to the TfR on the endothelial cells of capillaries in the brain, the molecule can be transferred into the CNS, just as iron is transferred (Figure 2). Human iduronate-2-sulfatase fused with an anti-transferrin receptor monoclonal antibody, Pabinafusp Alfa, was approved in Japan in 2021 after clinical trials showed no apparent side effects on iron metabolism [243,244,245]. Having achieved this, the focus has shifted to transportation efficiency through the BBB using the TfR. The factors determining this efficiency are numerous and complex and are not just limited to affinity to the receptor [246,247,248]. A clinical trial is currently underway using another fusion protein, DNL310, human iduronate-2-sulfatase fused to a modified Fc fragment, which can bind to the human TfR (ClinicalTrials.gov identifier: NCT04251026). Although the affinity of DNL310 for the TfR is lower than that of IgG:IDS, its transport efficiency into the brain was higher compared to IgG:IDS when 3 or 10 mg/kg of each drug was administered intravenously to mice expressing the human TfR apical domain [249]. The results of this clinical trial may provide better treatment options for patients in the future.

Insulin or iron are not the only substances that need to cross the BBB into the brain. In the future, more promising receptors for RMT may be identified. However, whatever the target receptor is, the potential effects of RMT on the original receptor functions must be observed closely. Also, as long as antibodies or similar proteins are used to bind to receptors, potential immune responses induced by the fusion protein should be carefully monitored [250].

In addition to the brain, avascular cartilage is another area where it is challenging to deliver enzymes. Skeletal deformity progresses with age, and the accumulation of GAGs at the growth plate inhibits bone growth [38]. To address this issue, enzymes conjugated with acidic amino acid oligopeptides with an affinity for hydroxyapatite in bone were used to study their effects by in vitro and in vivo experimental animal models. Tissue-nonspecific alkaline phosphatase (TNSALP) attached with six or eight residues of L-Asp at the *C*-terminus successfully increased the amount of enzyme in mouse bone up to 4-fold compared to untagged TNSALP [251]. This idea was later validated in MPS mouse models. The consequences of this tagged enzyme were studied in a mouse model of MPS VII by Montaño et al., who added six or eight aspartic acids to the N-terminus of human β-glucuronidase (GUS). The tagged enzyme intravenously administered had a five-times-longer blood clearance than an untagged enzyme and was successfully transported to the bone, bone marrow, and brain, where it reversed the storage condition [252]. When mice were given the tagged enzyme instead of the untagged enzyme, the serum HS and dermatan sulfate (DS) concentrations decreased more, even though none of the enzymes affected the cartilage [252]. Likewise, Tomatsu et al. attached six glutamic acids to the N-terminus of human N-acetylgalactosamine-6-sulfate sulfatase (GALNS; EC 3.1.6.4) and examined the effects of this artificial GALNS (E6-GALNS) in a mouse model of MPS IVA. E6-GALNS remained in mouse bone longer than regular GALNS, and after 24 weeks of infusion, light microscopy showed fewer vacuoles in bone and bone marrow. However, there was a poor improvement in the femur growth plate region, which is cartilage tissue [253]. To further examine the effects of acidic amino acid tags in-depth, Sawamoto et al. introduced an AAV8 vector expressing human GALNS with or without eight aspartic acids at the N-terminus under a liver-specific promoter in MPS IVA mice and then observed the mice for 12 weeks. Both vectors reduced the size of chondrocytes in the growth plate region compared to untreated mice; however, the size of chondrocytes in treated mice remained larger than in wild-type mice. There was no significant difference between the two vectors [254]. The continuous presence of high concentrations of each enzyme in the plasma probably resulted in increased penetration of each enzyme into the growth plate region and decreased KS levels in chondrocytes, even though the amount of enzyme that permeated into chondrocytes may not have been sufficient.

Based on the above, we conclude that acidic amino acid oligopeptides can deliver the necessary enzymes to the bone but remain insufficient to provide the enzymes to the cartilage, an essential organ for stable bone growth. However, this method could be applied to other enzyme replacement therapies by introducing negatively charged acidic amino acids into the enzyme so as not to affect the enzyme active site adversely. In the case of GALNS, the C-terminus is just near the positively charged active site of this enzyme and would be an inappropriate place to add negatively charged amino acids. Since the active site must be positively charged to attract negatively charged sulfate and carboxylate groups of GAGs, adding a negatively charged amino acid at the C-terminus of GALNS would prevent GAGs from approaching the active site, both physically and electrostatically (Figure 3).

### 4.2. Current Development of Gene Therapy for MPS

Gene therapy is a technology that aims to treat or cure diseases by modifying genes in the cells of organisms or by introducing therapeutic genes into cells. Vectors to deliver necessary DNA into a target cell fall into two categories: viral and non-viral. The latter is superior to the former regarding immunogenicity but is less able to transduce [255]. On the other hand, in the former case, if the patient has antibodies against the vector administered in vivo, the treatment will not be sufficiently effective, so the patient is excluded from clinical trials and is generally not eligible for gene therapy (for example, ClinicalTrials.gov identifier: NCT02702115, NCT03041324, NCT02716246, and NCT03173521). This section describes two viral vectors and one non-viral vector that are being investigated in the field of MPS.

#### 4.2.1. Adeno-Associated Virus Vector (AAV)

Adeno-associated virus (AAV) belongs to the genus *Dependroparvovirus* of the family *Parvoviridae* [256] and was first found as a contaminant in adenoviral preparations [257,258]. Its diameter is small, approximately 28 nm [259]. It is a nonreplicating, non-enveloped virus with a linear, single-stranded DNA (ssDNA) genome of approximately 4.8 kb [260,261]. This ssDNA, which encodes several genes, must first be converted to double-stranded DNA (dsDNA) in the host cell before its genetic information can be transcribed [262]. Recombinant AAV (rAAV) vectors lack AAV’s original genes and cannot replicate themselves. After an rAAV vector infects a cell, the ssDNA of rAAV will be converted to dsDNA to form a circular conformation and be able to persist as an episome in the nucleus of the transduced cell [263]. This circular episome is rarely incorporated into the host cell’s genomic DNA [264,265]. Even though episomes can replicate during cell division in some cases [266], in general, the effect of the transgene is progressively reduced, primarily when rAAV infects proliferating cells [260]. Thus, AAV is primarily effective against non-proliferating cells that do not frequently replicate, such as neurons, myocytes, or retinal cells. Although transgene integration into the genomic DNA of hepatocellular carcinoma has been observed in several studies in mouse models [267,268,269,270,271,272,273,274], it is not associated with hepatic genotoxicity in non-human primates or humans [265], suggesting that AAV is a safe viral vector for in vivo gene therapy. AAV has been the only viral vector used in in vivo gene therapy clinical trials for MPS [275,276].

As described in the previous paragraph, the production of dsDNA is thought to be necessary for AAV to express its transgenes and stay in the infected cells. Also, this is the rate-limiting step for efficient gene transfer in gene therapy [277]. However, this process can be eliminated by using self-complementary AAV (scAAV) vectors that contain transgenes already encoded in dsDNA, resulting in rapid and efficient effects both in vitro and in vivo [278]. Due to the upper limit of DNA size that an AAV capsid can hold, the size of the transgene that a scAAV can accommodate is about half that of a single-strand AAV (ssAAV) vector [279]; therefore, we cannot use this method for all genes. In the field of MPS, in vivo experiments in several mouse models confirmed its efficacy in MPS IIIA, where the causative gene *SGSH* is encoded in a cDNA within 1509 bp that can be stored in scAAV [280,281,282,283]. Currently, MPS IIIA is the only type employing scAAV in clinical trials (ClinicalTrials.gov identifier: NCT04088734, phase 1/2, terminated due to lack of efficacy in patients with cognitive development quotient (DQ) below 60; NCT02716246, phase 2/3, ongoing in patients with DQ 60 or above), although it would not be impossible for other types to use scAAV vectors, judging from the size of each responsible gene. For example, the cDNA of human *HYAL1* gene (NCBI reference sequence: NM_153281.2) is only 1308 bp, which is smaller and easier to pack into a scAAV vector than *SGSH*.

Eleven serotypes of AAV have been discovered in human cells and nonhuman primates [284]. AAV2 and AAV3, like other viruses, bind to HS in HSPGs on the cell membrane [285]. Positively charged basic amino acid residues constitute the HSPG-binding domain of AAV and attract negatively charged HS on the HSPG [285,286] (Figure 4A). Although HSPGs are not essential for these two viruses to infect cells [287,288], the affinity of the viral vector for HSPGs can be one of the critical determinants of its tissue tropism. Indeed, mutations in the HSPG-binding domain of the AAV2 capsid result in a phenotype of heparin-binding-deficient AAV, leading to increased cardiac transduction in vivo [286]. To obtain better tissue tropism and gene transfer efficiency, more than 90 serotypes have been artificially developed by modifying the capsid protein structure [289]. One of the candidates for future gene therapy for MPS is the bone-targeting vector. Carlos et al. introduced eight aspartic acids into the capsid protein of AAV2 immediately after the N-terminal amino acid of the VP2 protein and improved it so that the viral vector could be delivered efficiently to bones [290]. Aspartic acid is an acidic amino acid with negative electrostatic potential due to its carboxyl group. These negatively charged amino acids increased the affinity of the modified vector for positively charged hydroxyapatite in the bone matrix and increased vector genome copies in the bone of MPS IVA model mice [290] (Figure 4B). This bone-targeting vector could be a future treatment for skeletal dysplasia in congenital diseases, including MPS.

Generally, the percentage of children with antibodies to AAV gradually increases with age, and sensitization begins in early childhood [291]. Therefore, an early start of gene therapy might be preferable, but gene therapy enhances immune responses against the viral vector, preventing the second administration of the same viral vector [292]. On the other hand, the effects of gene therapy may gradually diminish as the patient grows, especially if the target cells are actively dividing and proliferating [264,293,294,295]. In summary, the younger the patient, the greater the probability that the AAV vector can be administered because the patient is less likely to be already sensitized to AAV, but also the greater the probability that a second gene therapy will be needed once the patient has grown up, even though the patient will inevitably produce antibodies to the administered AAV vector. Therefore, even if AAV gene therapy is approved after clinical trials are completed, there may be controversy over the optimal timing for administering viral vectors, especially when the target cells are actively proliferating cells, such as hepatocytes of young patients. 

As mentioned above, immunity is the most significant barrier to current in vivo gene therapy, and humoral immunity is the first barrier to overcome to deliver viral vectors to cells. Adopting AAV vectors with different serotypes to escape humoral immunity is one theoretical solution, yet designing another vector in a clinical setting poses practical difficulties. To address this challenging issue, Velazquez et al. proposed a method of reducing antibodies in mice with anti-AAV9 antibodies using rapamycin and prednisolone; however, their method also suppresses various lymphoid cells, which can lead to severe infections [296]. Meanwhile, Meliani et al. administered an AAV8 vector with nanoparticles encapsulating rapamycin to mice and nonhuman primates and immunosuppressed them, seemingly suppressing only the production of anti-AAV8 antibodies [297]. As a result, the same viral vectors were successfully administered for a second time [297]. However, there should still be concern about what happens if the patient is unfortunate enough to be infected with another microorganism when the viral vector is given together with the nanoparticles encapsulating rapamycin. Consequently, in in vivo AAV gene therapy, the optimal timing of vector administration and the feasibility of a second dose are still difficult problems to solve.

#### 4.2.2. Retroviral Vectors, including Lentiviral Vector

Retroviruses can be defined as viruses belonging to the *Retroviridae* family, including the *lentivirus* genus (https://talk.ictvonline.org/taxonomy/ (accessed on 12 November 2023)). These are a family of RNA viruses with a reverse transcriptase that creates a complementary DNA copy of the viral RNA, which is then integrated into DNA in a host cell. This family includes many vital pathogens, typically causing tumors or affecting the immune system, such as human immunodeficiency virus (HIV). Its virions are 80–100 nm in diameter and embed viral glycoproteins in the outer lipid envelope. The viral RNA is 7–12 kb in size, linear, single-stranded, non-segmented, and with positive polarity [298]. As mentioned above, transgenes can be incorporated into the host cell’s DNA, making retroviral vectors (RVs) suitable for treating proliferating cells, for example, ex vivo gene therapy targeting hematopoietic stem cells. Indeed, the first gene therapy clinical trial in 1990 was an ex vivo gene therapy targeting T lymphocytes and employed the RV vector [299]. When incorporated into the host’s genomic DNA, it can cause oncogenic mutations, a disadvantage of this vector [300,301,302]. Even though modifying promoters or enhancers on the long terminal repeat (LTR) of the RNA in the retroviral vector can alleviate the possibility of oncogenic mutations [303,304,305,306], this shortcoming is a significant hurdle in the application of this vector to in vivo gene therapy.

As mentioned in the former section, lentivirus (LV) is a genus belonging to the *Retroviridae* family. The main difference between lentiviruses and other retroviruses is their gene transfer capability. The former can infect both non-dividing and dividing cells, while the latter can only infect proliferating cells [307,308,309]. HIV is also included in the *lentivirus* genus.

In the field of MPS research, the first report using retroviral vectors was published in 1992, in which full-length human α-L-iduronidase (IDUA) cDNA was successfully administered to MPS I skin fibroblasts in vitro, ameliorating the intracellular accumulation of glycosaminoglycans observed by scintillation counting [310]. The expression level of IDUA differed among three promoters [310], suggesting the importance of the promoter, which is also of current interest in gene therapy [311]. Following in vivo animal experiments, in 2018, the first clinical trial of ex vivo LV gene therapy for MPS patients began, targeting hematopoietic stem cells from eight MPS I patients who could not find suitable donors for allogeneic HSCT (ClinicalTrials.gov identifier: NCT03488394, phase 1/2) [312], followed by the second one for MPS IIIA patients (ClinicalTrials.gov identifier: NCT04201405, phase 1/2). Positive results from these clinical trials would encourage patients struggling to find a donor for an allogeneic HSCT. Also, in the case of ex vivo gene therapy, the therapeutic gene can be introduced in vitro, where the immune cells and antibodies in patients have been removed so that the immune response is not an issue when introducing the gene, which is an advantage over in vivo gene therapy. However, because we need to manipulate cells in vitro to edit the genome, the types of cells we can treat directly are practically limited.

#### 4.2.3. Sleeping Beauty Transposon System

This system, named for its ability to reawaken “sleeping” DNA sequences in the genome, uses a type of transposon, which is a segment of DNA that can move around to different positions within the genome [313,314]. In this approach, the Sleeping Beauty transposon carries a therapeutic gene into a patient’s cells [315]. Once inside the cell, the transposon system integrates the therapeutic gene into the patient’s genome [315]. This is achieved through the activity of transposase, which is specifically provided by this system. The transposase cuts the DNA and inserts the transposon, along with the therapeutic gene, into a new location in the genome [315]. The key advantage of this system would be its precision and control in integrating genes compared to retroviral vectors. Retroviral vectors can insert genes at random locations, disrupting important genes or regulatory regions in the DNA and leading to harmful side effects such as oncogenicity [300,301,302]. In contrast, the Sleeping Beauty system can be designed to target more specific sites in the genome, thereby reducing the risk of unintended genetic disruptions. This system has shown promise in preclinical trials for various diseases, including cancer and genetic disorders [314,315,316,317,318]. In vivo MPS I and VII animal model experiments demonstrated positive results with this method [319,320,321,322]. On the other hand, for ex vivo gene therapy, this method has been studied mainly for the generation of chimeric antigen receptor T (CAR-T) cells [323,324]. In combination with these positive results, a phase 1 clinical trial (NCT05682144) was initiated, in which ex vivo gene therapy using this transposon system was applied to autologous plasmablasts from MPS I patients. Selecting the B-cell lineage rather than hematopoietic stem cells as a treatment target would be a prudent strategy to further reduce the risk of oncogenicity.

### 4.3. Immunomodulatory Drugs Tested in MPS Models (Figure 1)

#### 4.3.1. Monoclonal Antibody against Tumor Necrosis Factor-Alpha (TNF-α)

Various cell types, including some non-immune cells, produce TNF-α. One of the signaling pathways that can activate the secretion of TNF-α is the NF-κB pathway [325]. This pathway is often activated downstream of Toll-like receptors (TLRs), essential for the innate immune response [325,326]. 

Simonaro et al. (2010) and Eliyahu et al. (2011) researched the anti-TNF-α drugs Infliximab and CNTO1081, respectively [181,327]. They used an MPS animal model to examine its efficiency. Their findings highlighted the key role of the TLR/TNF-α inflammatory pathway in MPS skeletal issues. They also showed that combining ERT with anti-TNF-α drugs improved outcomes and supported the use of TNF-α as a potential biomarker for MPS.

Fast-forwarding to 2017, Polgreen et al. conducted a 32-week, randomized, double-blind, placebo-controlled, crossover study on Adalimumab, a human monoclonal antibody that blocks TNF-α [328]. This study involved MPS I and II patients and used the clinical trial identifiers NCT02437253 and NCT03153319. Two patients, one with MPS I and the other with MPS II, completed this trial. Notably, no severe side effects were reported. The preliminary data suggested that Adalimumab might help alleviate pain and improve both physical and neurogenic functions in children suffering from MPS I or II.

The large size of monoclonal antibodies usually means they have limited access to the central nervous system (CNS), as suggested by Müller-Miny et al. [329]. However, their ability to alter the immune system outside of the CNS can indirectly affect inflammation within it, as Torres-Acosta et al. noted that reducing TNF-α in the body led to positive effects in the CNS in a mouse model studying Alzheimer’s disease [330]. Reducing the level of inflammatory cytokines produced outside the CNS may lower the amount that can traverse the BBB to some extent [331]. This suggests that the treatment could also benefit MPS patients with nervous system issues.

#### 4.3.2. Interleukin-1 Receptor Antagonist: Anakinra

Anakinra is a recombinant, nonglycosylated form of the human interleukin-1 receptor antagonist. It consists of 153 amino acids and has a molecular weight of 17.3 kDa. It is produced using an *E. coli* bacterial expression system (https://www.accessdata.fda.gov/drugsatfda_docs/label/2012/103950s5136lbl.pdf (accessed on 25 November 2023)).

Anakinra has undergone a phase II/III clinical trial in 20 patients with MPS III (NCT04018755) and can be effective for neurocognitive involvement. Thanks to its small size, this molecule crosses the human brain-like endothelium monolayer at 4.7 times the rate of the monoclonal antibodies examined, according to Sjöström et al. [332]. As a result, anakinra may reach clinically meaningful levels in the CNS, supporting its value in treating neuroinflammatory conditions. It is currently being used in another clinical trial to suppress secondary brain injury after spontaneous intracerebral hemorrhage [333]. This accessibility to the CNS would also be advantageous for treating CNS symptoms in MPS.

#### 4.3.3. Anti-Oxidant Drugs

As discussed in the previous sections, ROS promotes inflammation via the synthesis of inflammasome. Reducing oxidative stress and ROS is vital to alleviate excessive immune response.

Resveratrol, recognized as a natural phenolic compound and a phytoalexin, emerges as a promising agent due to its multifaceted role in stimulating autophagy. It works as an antioxidant, an anti-inflammatory agent, and a regulator of autophagy [334]. In a Drosophila melanogaster model of MPS VII, the administration of resveratrol resulted in behavioral enhancement and successful penetration through the BBB [335]. The safety and tolerability of resveratrol have been confirmed in a range of animal models and human studies, including those focusing on neurodegenerative diseases. The likelihood of potential adverse effects is deemed minimal [334]. Rintz et al. demonstrated that the administration of 50 mg/kg/day of resveratrol for 12 weeks or longer improved neurological symptoms, autophagy in the brain and liver, and decreased urinary GAG levels in an MPS IIIB mouse model [336]. Given the evidence that oxidative stress and compromised autophagy are implicated in the augmentation of inflammatory processes in a spectrum of lysosomal disorders, it would be prudent to investigate the efficacy of resveratrol in various forms of MPS.

Coenzyme Q10, also known as ubiquinone, is a compound that plays a crucial role in the mitochondrial electron transport chain, which is vital for the production of ATP. It also serves as an antioxidant, helping protect cells from damage by neutralizing free radicals. These potentially harmful molecules can lead to oxidative stress and damage cellular components such as proteins, lipids, and DNA. The antioxidant properties of coenzyme Q10 are critical in maintaining the integrity of cell membranes and lipoproteins. Montero et al. investigated its concentration in the plasma from 44 MPS patients (6 MPS I, 2 MPS II, 16 MPS IIIA, 5 MPS IIIB, 7 MPS IIIC, 4 MPS IV, 3 MPS VI, and 1 MPS VII patients). Except for Hurler–Scheie syndrome and MPS VI, all had lower coenzyme Q10 concentrations than controls [337]. Moreover, Jacques et al. showed that the administration of 5–10 μM of coenzyme Q10 decreased oxidative stress in IDS-deficient HEK 293 cells in vitro [338], suggesting that the evaluation and supplementation of coenzyme Q10 could be beneficial to MPS patients.

#### 4.3.4. Cathepsin B Inhibitor: CA-074

Cathepsin B is one of the key factors in amplifying the inflammatory pathway in LSD [153,154,339]. Several animal models have shown an elevated expression and/or activity of cathepsin B in the brain (MPS I, II, IIIA, and VII), cardiovascular system (MPS I and VII), and bone in MPS VII [340]. Baldo et al. suggested a possible relationship between increased cathepsin B activity and cardiovascular symptoms [341], which was subsequently confirmed by Gonzalez et al. using the cathepsin B inhibitor Ca-074 in an MPS I mouse model [342]. Considering the ubiquitous involvement of cathepsin B in LSDs, the therapeutic potential of Ca-074 warrants investigation across a spectrum of these disorders.

#### 4.3.5. Pentosan Polysulfate

Pentosan polysulfate (PPS, C_10_H_18_O_21_S_4_, PubChem CID 37720) is a sulfated pentosyl polysaccharide with heparin-like properties. This molecule can promote the proliferation and chondrogenic differentiation of adult human bone-marrow-derived mesenchymal progenitor cells [343]. PPS regulates the production of cytokines and inflammatory mediators, and it is also an anti-tumor agent that encourages the proliferation and differentiation of mesenchymal stem cells, as well as the development of progenitor cell lineages that are useful in tactics designed to repair cartilage in osteoarthritis [344].

Earlier subcutaneous PPS administration demonstrated a reduction in neuroinflammation and neurodegeneration in the brain of an MPS IIIA mouse [345]. Given its molecular size, oral administration would also be possible; however, animal models showed more effective results with subcutaneous injection than oral administration, reducing cytokine levels in serum and CSF [346,347]. Based on these positive preclinical results, clinical studies were initiated and demonstrated a reduction in urinary GAG excretion, improvement in skeletal pathology and symptoms in four MPS I patients [348], and decreased inflammatory cytokines in three MPS II patients [349]. Detailed molecular mechanisms of this molecule for MPS remain unclear, but the observations above support additional clinical studies.

### 4.4. Immunomodulatory Drugs Not Yet Tested in MPS Models (Figure 1)

Upon meticulously reviewing the existing scholarly literature, we recognize a dearth of empirical research incorporating the MPS model with the reagents delineated herein. Nonetheless, considering the pharmacodynamics of these compounds, they could potentially be considered as promising candidate agents for MPS and other LSDs.

#### 4.4.1. Anti-Oxidant Drug

VI-16 is a synthetic flavonoid compound that was designed to interact with the NLRP3 inflammasome. In addition to reducing mitochondrial ROS generation, this molecule can inhibit the binding of TXNIP to NLRP3 and intervene in the activation of NLRP3 inflammasome, as shown in colitis of mice induced by dextran sulfate sodium (DSS) [350]. 

#### 4.4.2. Caspase-1 Inhibitor

Pro-caspase-1 constitutes an essential element of the NLRP3 inflammasome complex. After the orchestration of the NLRP3 inflammasome, pro-caspase-1 undergoes proteolytic cleavage to yield its active form, caspase-1 [351]. The resultant active caspase-1 then catalyzes the conversion of pro-IL-1β and pro-IL-18 into their active cytokine counterparts, IL-1β and IL-18, respectively [351,352]. Consequently, the inhibition of caspase-1 may diminish these cytokines, which could ameliorate inflammation. Within this classification, agents such as belnacasan (VX-765), pralnacasan (VX-740), and parthenolide exemplify the representative reagents [353,354,355,356]. Belnacasan was evaluated in a phase 2 clinical trial in treating COVID-19 (NCT05164120).

#### 4.4.3. Gasdermin D (GSDMD) Inhibitor: Disulfiram

Activated caspase-1 cleaves gasdermin D (GSDMD), specifically at its central linker, separating the N-terminal domain from the C-terminal domain [357]. The N-terminal fragment of GSDMD possesses an intrinsic pore-forming activity and inserts into the plasma membrane to form oligomeric pores [357]. These pores disrupt cellular ion gradients and osmotic stability, leading to cell swelling and eventual rupture, a process termed pyroptosis, an inflammatory form of programmed cell death [358]. The formation of pores in the plasma membrane by GSDMD contributes to cell death and facilitates the release of the mature cytokines IL-1β and IL-18 into the extracellular milieu, thereby perpetuating and amplifying the inflammatory response [358]. This pathway underscores a key mechanism through which the innate immune system responds to infections and tissue damage, linking the cellular death process to the secretion of inflammatory cytokines and the promotion of inflammation.

Disulfiram is a drug that is primarily used to support the treatment of chronic alcoholism by producing an acute sensitivity to ethanol [359]. Interestingly, research has shown that disulfiram can also inhibit the formation of pores by GSDMD, thereby inhibiting pyroptosis [360]. Disulfiram can bind covalently to Cys191 within the N-terminal domain of GSDMD, blocking its ability to oligomerize, which is required for forming pores in the cell membrane [360]. Thus, by inhibiting the pore-forming activity of GSDMD, disulfiram can reduce the release of inflammatory cytokines and other inflammatory mediators that occur during pyroptosis, thereby dampening the inflammatory response [360,361,362]. This inhibition offers a potential therapeutic target for treating diseases that involve excessive inflammation, such as Lyme disease or COVID-19, for which disulfiram has already been tested in clinical trials (NCT03891667, NCT04485130, NCT04594343). This substantial prior clinical experience would make it feasible and practical to test disulfiram in MPS.

#### 4.4.4. Inhibitor of TLR4 Signaling: TAK-242

TAK-242 (resatorvid) binds to the Toll/IL-1 receptor (TIR) domain of TLR4 via Cys747 and then interferes with protein–protein interactions between TLR4 and its adaptor molecules [363,364]. Several animal models have demonstrated its anti-inflammatory effects [365,366,367]. Unfortunately, it failed to suppress cytokine levels in phase 3 clinical trials for sepsis (NCT00633477, NCT00143611) [368]. Nevertheless, considering that the inflammation associated with MPS is chronic in nature, as opposed to the acute inflammatory response observed in sepsis, the potential efficacy of TAK-242 in mitigating MPS symptoms may still constitute a plausible hypothesis.

#### 4.4.5. Reagents Capable of Inhibiting NLRP3 Inflammasome

As we mentioned in the previous sections, several possible reagents can be useful to suppress excessive inflammation in MPS, and as far as we can review the past literature, most related inflammatory pathways must go through the inflammasome complex (Figure 1). Thus, inhibition of NLRP3 inflammasome itself seems to be effective as well. 

Two oral NLRP3 inhibitors produced by Inflazome, inzomelid (clinical trials identifier: NCT04086602) and Somalix, are intended for the treatment of cryopyrin-associated periodic syndrome (CAPS) and cardiovascular disease, respectively [369]. The phase I clinical trial of NT-0796, an NLRP3 inflammasome inhibitor for neuroinflammatory disorders, has been completed and demonstrated the brain penetration of the compound [370]. This category includes many additional compounds [369,371], but to the best of our knowledge, none have been evaluated for treating LSD models. In other words, there could still be many potential options for MPS patients in this category.

#### 4.4.6. NF-κB (Nuclear Factor Kappa-Light-Chain-Enhancer of Activated B Cells) Inhibitor

NF-κB plays a key role in regulating immune responses to infection. Dysregulation of NF-κB has been linked to cancer, inflammatory and autoimmune diseases, septic shock, viral infection, and improper immune development [372]. When activated, NF-κB translocates to the nucleus of the cell, where it turns on the expression of specific genes that are involved in the body’s inflammatory response, for example, NLRP3, pro-IL-1β, pro-IL-18, TNF-α, and IL-6 [189,190,373].

BAY 11-7082 (https://pubchem.ncbi.nlm.nih.gov/compound/E_-3-Tosylacrylonitrile (accessed on 26 November 2023)) prevents the translocation of NF-κB to the nucleus, thereby inhibiting its ability to activate inflammatory gene transcription [374,375]. It can also inhibit NLRP3 inflammasome independently of NF-κB [376]. Its efficacy has been demonstrated in several animal models [377,378,379,380]. Due to its small molecular weight (207.25 Da), this molecule can easily cross the BBB [381], making it attractive for alleviating CNS symptoms of MPS.

### 4.5. Other Potential Therapies in the Future

Concerning substrate reduction therapy, miglustat, an inhibitor of glucosylceramide synthase, was orally administered to 25 patients with MPS III in a clinical trial. Unfortunately, miglustat, despite its ability to cross the blood–brain barrier, did not improve or stabilize behavioral problems in MPS III patients [382]. 

A β-D-xyloside analog administered orally, odiparcil (CAS#137215-12-4), may not be included in substrate reduction therapy for LSD in the typical sense. However, this reagent can reduce GAG accumulation by increasing the urinary excretion of GAGs [383]. The attachment of a galactosyl molecule to D-xylose linked to a serine residue in a proteoglycan is the first step in the elongation of a polysaccharide chain of CS, DS, and HS, catalyzed by galactosyltransferase I (β4GalT7) [384,385,386]. However, odiparcil can interfere with this reaction because odiparcil is a more reactive substrate for β4GalT7 than D-xylose [387,388]. As a result, the polysaccharide chain will be extended on odiparcil rather than on a proteoglycan. This incorrectly elongated polysaccharide chain will be released from the cells without undergoing lysosomal degradation and eventually excreted in the urine [383,389]. Therefore, odiparcil can “theoretically” reduce three GAGs: CS, DS, and HS. Although not approved, this reagent was originally developed as an anticoagulant [390,391]. Subsequently, its role in reducing accumulated GAGs was reconsidered, and a phase 2 clinical trial for MPS VI patients was completed [392]. Participants in this study were between 16 and 64 years of age, so, as Guffon et al. pointed out, further evaluation of odiparcil in younger patients should be considered. In addition, given the molecular mechanisms of odiparcil, other types of MPS, particularly MPS VII, may also be a suitable target for this reagent.

## 5. Concluding Remarks

In this review, we have described the pathophysiology of MPS from a molecular biological aspect to obtain an idea of which molecules could be aimed at to further improve the prognosis of patients with MPS. ERT has already improved the quality of life (QOL) of MPS patients to some extent; however, it still leaves burdens for patients, such as frequent injections, financial concerns, and difficulty in treating the CNS and bone. The introduction of gene therapy will further improve the QOL, but financial concerns may be a bigger problem. Also, patients with late diagnosis may not be able to receive the benefits of gene therapy, as younger patients with little progression of symptoms would be preferable candidates for future gene therapy. Even for such patients, we believe there would still be possibilities to improve their symptoms, as there are a variety of candidate molecules to be suppressed in their activated inflammatory pathways.

We hope that further research and novel findings in this area will benefit all MPS patients in the future, irrespective of their medical history or financial situation.

## Figures and Tables

**Figure 2 ijms-25-01113-f002:**
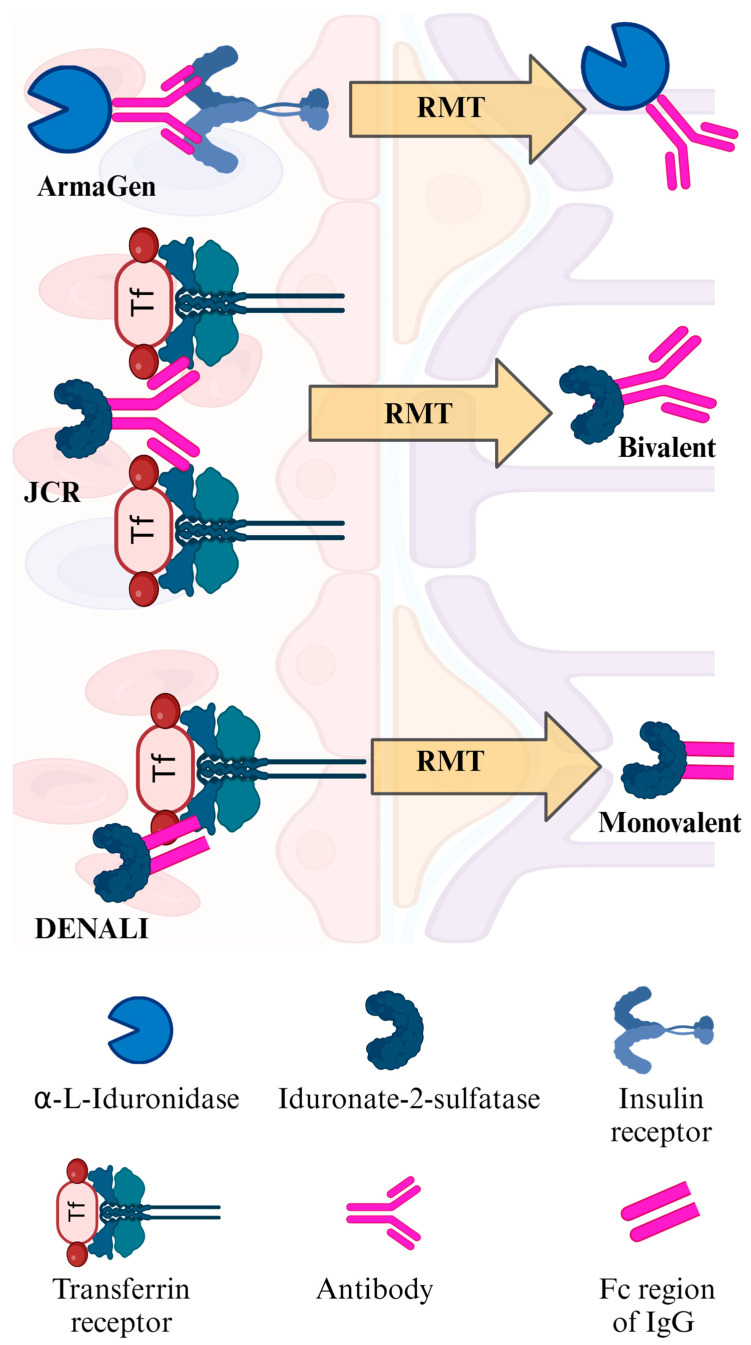
Receptor-mediated transcytosis across the blood–brain barrier. Receptor-mediated transcytosis (RMT) (molecular Trojan horse) mediates the transfer of enzymes tagged with monoclonal antibodies or ligands targeting specific receptors on the surface of the endothelial cells of the blood–brain barrier. The modality of monoclonal antibodies or ligands targeting transferrin receptors differs depending on the gene product. ArmaGen, Inc. (Calabasas, CA, USA) and JCR Pharmaceuticals Co., Ltd. (Ashiya, Japan)—bivalent (enzyme fused with the whole antibody). DENALI Therapeutics (South San Francisco, CA, USA)—monovalent (enzyme fused with the Fc region of IgG). This figure was created with Biorender.com (accessed on 1 December 2023).

**Figure 3 ijms-25-01113-f003:**
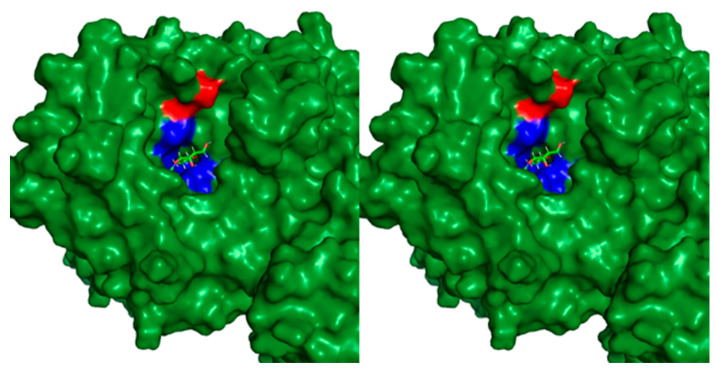
Three-dimensional structural model representing the surface of GALNS (Protein Data Bank ID: 4FDJ), cross-eyed stereo view. The blue area is the surface of the active site consisting of Asp39, Asp40, Arg83, Tyr108, Lys140, His142, His236, Asp288, Asn289, Lys310, and dihydroxyalanine 79. The red area is the surface of Trp520. These two areas are in contact. This model does not determine the location of two amino acid residues, Ser 521 and His 522, at the C-terminus. The molecule depicted in the stick model is N-acetylgalactosamine. This figure was created with PyMOL Version 2.5.2.

**Figure 4 ijms-25-01113-f004:**
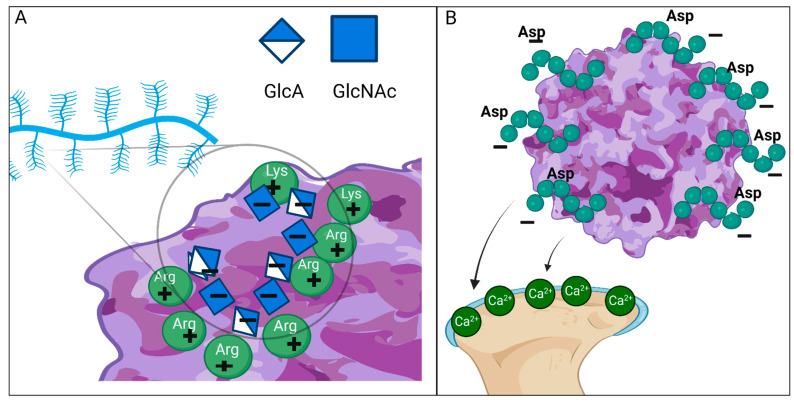
Electrostatic potential of AAV capsid protein and its affinity to targets. (**A**) Heparan sulfate proteoglycan (HSPG) is negatively charged with disaccharide units of β-D-glucuronic acid (GlcA) and N-acetylglucosamine (GlcNAc). It has an affinity towards the positively charged HSPG-binding domain of AAV capsid, which consists of basic amino acids such as Arg and Lys. (**B**) Modification of VP1 and VP2 capsid proteins with negatively charged amino acid (Asp) on AAV increases the affinity of the modified vector for positively charged calcium hydroxyapatite in bone matrix. This figure was created with Biorender.com (accessed on 1 December 2023).

**Table 1 ijms-25-01113-t001:** Types of MPS and storage material [17,48,49,50,51].

Disease	Impaired Enzyme	Gene	Primary Storage Material	Symptoms
Skeletal disease, soft tissue storage, and CNS symptoms				
MPS I (Hurler syndrome)	α-L-Iduronidase	*IDUA*	DS, HS	Dysostosis multiplex, short stature, short neck and trunk, kyphoscoliosis, rigidity of joints, micrognathia, coarse facial features, macroglossia, retinal degeneration, corneal clouding, cardiac valvular disease, cardiomyopathy, hepatosplenomegaly, cognitive impairment, developmental delay
MPS II (Hunter syndrome)	Iduronate-2-sulfatase	*IDS*	DS, HS	Dysostosis multiplex, short stature, short neck and trunk, kyphoscoliosis, rigidity of joints, coarse facial features, cardiac valvular disease, no corneal clouding, retinal degeneration, hepatosplenomegaly, cognitive impairment, developmental delay
MPS VII (Sly syndrome)	β-Glucuronidase	*GUSB*	HS, C4S and C6S DS	Dysostosis multiplex, short stature, short neck and trunk, kyphoscoliosis, rigidity of joints, coarse facial features, hydrops fetalis, hepatosplenomegaly, corneal clouding, cognitive impairment, developmental delay
Mainly CNS involvement with less skeletal and soft-tissue disease				
MPS IIIA (Sanfilippo syndrome A)	N-sulfoglucosamine sulfohydrolase	*SGSH*	HS	Developmental delay, cognitive impairment, hyperactivity, spasticity, motor dysfunction
MPS IIIB (Sanfilippo syndrome B)	N-acetyl-α-glucosaminidase	*NAGLU*	HS	
MPS IIIC (Sanfilippo syndrome C)	Acetyl-CoA:α-glucosaminide N-acetyltransferase	*HGSNAT*	HS	
MPS IIID (Sanfilippo syndrome D)	Glucosamine (N-acetyl)-6-sulfatase	*GNS*	HS	
Mainly skeletal, cartilage, and ligament disease; no CNS symptoms				
MPS IVA (Morquio syndrome A)	Galactose/N-acetylgalactosamine-6-sulfatase	*GALNS*	KS, C6S	Dysostosis multiplex, short stature, short neck and trunk, pectus carinatum, kyphoscoliosis, genu valgum, motor dysfunction, hyperlaxity of joints, cardiac valvular disease, corneal clouding, narrowing trachea: MPS IVA has more severe symptoms.
MPS IVB (Morquio syndrome B)	β-Galactosidase	*GLB1*	KS	
Skeletal disease and soft tissue storage; no CNS symptoms				
MPS VI (Maroteaux–Lamy syndrome)	N-acetylgalactosamine-4-sulfatase	*ARSB*	DS, C4S	Dysostosis multiplex, short stature, short neck and trunk, kyphoscoliosis, motor dysfunction, cardiac valvular disease, corneal clouding
MPS X	Arylsulfatase K	*ARSK*	DS, HS, KS, CS	Dysostosis multiplex, short stature, coarse facial features, mild lens and vitreous opacity, cardiac valvular disease
Mainly soft tissues around joints, with no CNS symptoms				
MPS IX (Natowicz syndrome)	Hyaluronidase	*HYAL1*	HA	Large-joint effusion, periarticular soft-tissue masses, mild facial changes, short stature
Unclassified				
MPS-plus syndrome	Target protein: vacuolar protein sorting-associated protein 33A	*VPS33A*	DS, HS	Increased prenatal thickness of the fetus, coarse facial features, dysostosis multiplex, rigidity of joints, congenital heart defects, anemia, hepatomegaly, nephromegaly, proteinuria, psychomotor delay

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
