# Peer review of "Molecular Mechanisms in Pathophysiology of Mucopolysaccharidosis and Prospects for Innovative Therapy"

_ijms, 2024, doi:10.3390/ijms25021113_

Round 1

Reviewer 1 Report

Comments and Suggestions for Authors

Ago et al. comprehensively reviewed current understanding of pathophysiology of MPS and its therapy. The article is informative for readers of IJMS especially for physicians and researchers in this field. The reviewer has only few minor comments.

1. Line 464:

It may be better to cite Figure 2 after the next phrase. “By binding… is transferred (Figure 2).”

2. Lines 473-476:

The authors mentioned that transport efficacy of DNL310 into the brain is higher than Pabinafusp Alfa by citing the article 245. In this paper, however, the Denali team used an IgG-type anti-transferrin receptor antibody, which is not identical to Pabinafusp Alfa, as the reference material for comparing their ETV:IDS (DNL310). The explanation is not accurate. "Pabinafusp Alfa" should be changed to "IgG:IDS".

Author Response

Author's Reply to the Review Report (Reviewer 1)

Ago et al. comprehensively reviewed current understanding of pathophysiology of MPS and its therapy. The article is informative for readers of IJMS especially for physicians and researchers in this field. The reviewer has only few minor comments.

Response: We appreciate your review of our manuscript.

  1. Line 464:

It may be better to cite Figure 2 after the next phrase. “By binding… is transferred (Figure 2).”

Response: Thank you for your suggestion. We have cited Figure 2 as you instructed.

  1. Lines 473-476:

The authors mentioned that transport efficacy of DNL310 into the brain is higher than Pabinafusp Alfa by citing the article 245. In this paper, however, the Denali team used an IgG-type anti-transferrin receptor antibody, which is not identical to Pabinafusp Alfa, as the reference material for comparing their ETV:IDS (DNL310). The explanation is not accurate. "Pabinafusp Alfa" should be changed to "IgG:IDS".

Response: Thank you for your incisive remarks. We have changed " Pabinafusp Alfa " to "IgG:IDS" in the sentence you mentioned.

Reviewer 2 Report

Comments and Suggestions for Authors

Molecular Mechanisms in Pathophysiology of Mucopolysac-2 charidosis and Prospects for Innovative Therapy. Below are presented my suggestions in blue.

Lane 18: progress irreversibly to:1)

Lane 43: the authors indicate that the patients with MPS do not produce one of the lysosomal enzymes necessary to break down these GAGs or generate enzymes that do not work appropriately. It is possible that a little more could be explained about the enzyme failures associated with this disease.

Lane 49. include the following sentence: Each subtype of MPS is associated with the deficiency of a particular enzyme, leading to the accumulation of specific types of GAGs as wil be described more below.

Please: It is important to note that each MSP III, IV etc. are known as Sanfilippo Syndrome (Subtypes xxxx) and Morquio Syndrome (Subtypes A and B). place it somewhere in the text.

Lane 164. They also confirmed in vitro by They also confirmed through in vitro studies.

Lane 166: To explain a little better how DA accumulation occurs in fibroblasts and not only place the reference.

Finally, the manuscript is well written, clearly describing how each GAG component influences the pathology. The figure is clear. Also, the article shows possible targets of action for therapies in a robust manner. The article is interesting and should be published.

 Can be accepted with minor corrections.

Comments on the Quality of English Language

The paper is well written, there are small mistakes, but I ´m not a native English speaker.

Author Response

Author's Reply to the Review Report (Reviewer 2)

Molecular Mechanisms in Pathophysiology of Mucopolysaccharidosis and Prospects for Innovative Therapy.

Response: We appreciate your review of our manuscript.

Lane 18: progress irreversibly to:1)

Response: Thank you for your correction, We added “to” as you instructed.

Lane 43: the authors indicate that the patients with MPS do not produce one of the lysosomal enzymes necessary to break down these GAGs or generate enzymes that do not work appropriately. It is possible that a little more could be explained about the enzyme failures associated with this disease.

Response: Thank you for your suggestion. We have added the following additional words to explain more.

 “Patients with MPS do not produce one of the lysosomal enzymes necessary to break down these GAGs or generate enzymes that do not work appropriately [11], depending on the type of genetic mutations on each responsible gene, e.g., nonsense mutation, missense mutation, deletion, etc.”

Lane 49. include the following sentence: Each subtype of MPS is associated with the deficiency of a particular enzyme, leading to the accumulation of specific types of GAGs as will be described more below.

Response: We appreciate your suggestion. We have added this sentence to the recommended position with a minor modification as follows.

“Each subtype of MPS is associated with the deficiency of a particular enzyme, leading to the accumulation of specific types of GAGs, as described below.”

Please: It is important to note that each MPS III, IV etc. are known as Sanfilippo Syndrome (Subtypes xxxx) and Morquio Syndrome (Subtypes A and B). place it somewhere in the text.

Response: Thank you for your suggestion. We added the names in Table 1 except MPS X which is unnamed.

Lane 164. They also confirmed in vitro by “They also confirmed through in vitro studies.”

Response: We appreciate your correction. We have corrected the pointed part as you instructed.

Lane 166: To explain a little better how DS accumulation occurs in fibroblasts and not only place the reference.

Response: Thank you for your suggestion. We have added more information.

“Using fibroblasts with MPS III A, B, C, and D patients, Lamanna et al. found an increase in intracellular dermatan sulfate (DS), not just HS. They also confirmed through in vitro studies that HS inhibits iduronate 2-sulfatase, the enzyme deficient in MPS II [75]. This finding was consistent with previous results from a diagnostic approach, where unexplained elevations of DS were detected in MPS III patient serum and urine [76].”

Finally, the manuscript is well written, clearly describing how each GAG component influences the pathology. The figure is clear. Also, the article shows possible targets of action for therapies in a robust manner. The article is interesting and should be published.

 Can be accepted with minor corrections.

Response: We appreciate your positive evaluation; thank you so much.

Additional refs

  1. Tomatsu, S.; Montaño, A. M.; Oguma, T.; Dung, V. C.; Oikawa, H.; Gutiérrez, M. L.; Yamaguchi, S.; Suzuki, Y.; Fukushi, M.; Barrera, L. A.; et al. Validation of Disaccharide Compositions Derived from Dermatan Sulfate and Heparan Sulfate in Mucopolysaccharidoses and Mucolipidoses II and III by Tandem Mass Spectrometry. Mol. Genet. Metab., 2010, 99 (2), 124–131. https://doi.org/10.1016/j.ymgme.2009.10.001.

Reviewer 3 Report

Comments and Suggestions for Authors

In this review Ago and colleagues discuss the molecular mechanisms underlying the effects of Mucopolysaccharidosis and innovative therapeutical strategies to alleviate the symptoms or treat the disease.

Overall, I consider that this review provides a general view on MPSs with a good level of detail, while not extending its content too long. The manuscript is easy to read, it is well-organised, and provides a good landscape over this field of study. I would like to point out a few minor issues that should be addressed:

1)     Line 47. The authors refer “MPS types” but a table explaining the characteristics of each MPS type is only displayed later. A citation to table 1 would fit well in this line. Also, it would be beneficial to shortly introduce the classification of MPSs before mentioning the different types for the first time.

2)     Lines 120-121. Since MPS II is a X-linked disease, its prevalence must differ between men and women. MPS II must also be more prevalent than the other autosomal recessive MPSs. This information could be added here.

3)     Lines 183-185. The “toxic proteinopathy hypothesis” has been challenged recently (1) after it was found that the seminal paper showing the link between amyloid beta accumulation and neurodegeneration (2) was based on fabricated data (3). I recommend caution when taking these phenomena as absolute truths in neurodegeneration.

4)     Lines 428-438. These paragraphs should be included in the legend of figure 1.

5)     Lines 480-488. These paragraphs should be included in the legend of figure 2.

6)     Lines 539-544. These paragraphs should be included in the legend of figure 3.

7)     Lines 586-587. The authors employ the plural although just one Clinical Trial is cited. As the authors state in section 4.2, only one clinical trial is discussed in each subsection 4.2.x, and that is perfectly reasonable. However, in these lines, it would be interesting to cite other ongoing trials, if any. Besides that, it would be beneficial to include more details about the clinical trial discussed by the authors, specially concerning the present phase of the trial and main results thus far.

8)     Lines 642-649. These paragraphs should be included in the legend of figure 4.

9)     Lines 681-682. Are these Clinical Trials parts of the same trial? Again, it would be beneficial to include information about the trials’ phase in the text.

Bibliography:

1.            Espay AJ, Herrup K, Daly T. Chapter 10 - Finding the falsification threshold of the toxic proteinopathy hypothesis in neurodegeneration. In: Espay AJ, editor. Handbook of Clinical Neurology. 192: Elsevier; 2023. p. 143-54. https://doi.org/10.1016/B978-0-323-85538-9.00008-0

2.            Lesné S, Koh MT, Kotilinek L, Kayed R, Glabe CG, Yang A, et al. A specific amyloid-β protein assembly in the brain impairs memory. Nature. 2006;440(7082):352-7. 10.1038/nature04533

3.            Piller C. Blots on a field? Science (New York, NY). 2022;377(6604):358-63. 10.1126/science.add9993

Author Response

Author's Reply to the Review Report (Reviewer 3)

In this review, Ago and colleagues discuss the molecular mechanisms underlying the effects of mucopolysaccharidosis and innovative therapeutic strategies to alleviate the symptoms or treat the disease.

Overall, I consider that this review provides a general view of MPSs with a good level of detail while not extending its content too long. The manuscript is easy to read, it is well-organized, and provides a good landscape over this field of study. I would like to point out a few minor issues that should be addressed:

Response: We appreciate your positive review of our manuscript.

1)     Line 47. The authors refer “MPS types,” but a table explaining the characteristics of each MPS type is only displayed later. A citation to table 1 would fit well in this line. Also, it would be beneficial to shortly introduce the classification of MPSs before mentioning the different types for the first time.

Response: We appreciate your suggestion. We have modified the pointed position as follows.

“To date, eight distinct clinical types and subtypes of MPS III (A, B, C, D) and IV (A, B) have been identified, and 12 diseases were classified as MPS (Table 1). Most MPS types, except MPS IV and VI, have primary central nervous system (CNS) involvement, while most patients, except MPS III, have progressive systemic skeletal dysplasia [13]. Each subtype of MPS is associated with the deficiency of a particular enzyme, leading to the accumulation of specific types of GAGs, as described below. In 2014, Dr. Gurinova et al. [14] reported a novel disease of impaired GAG metabolism without deficiency of known lysosomal enzymes: mucopolysaccharidosis-plus syndrome (MPSPS: OMIM #617303) [15,16]. MPSPS is an autosomal recessive multisystem disorder caused by a specific mutation p.R498W in the vacuolar protein sorting-associated protein 33A (VPS33A) gene. The name of the disease, MPSPS, means that in addition to typical symptoms for conventional MPS I, patients developed other features such as congenital heart defects and renal and hematopoietic disorders. It remains unknown how missense mutation p.R498W in VPS33A causes the accumulation of GAGs. Detailed mechanisms and disease pathophysiology remain to be elucidated [17].”

2)     Lines 120-121. Since MPS II is an X-linked disease, its prevalence must differ between men and women. MPS II must also be more prevalent than the other autosomal recessive MPSs. This information could be added here.

Response: Thank you for your suggestion. I have added descriptions according to your instructions as follows.

“Except for MPS II, which is X-linked and therefore has a relatively high prevalence compared to other types of MPS, especially in males, all MPS are autosomal recessive diseases [47].”

3)     Lines 183-185. The “toxic proteinopathy hypothesis” has been challenged recently (1) after it was found that the seminal paper showing the link between amyloid beta accumulation and neurodegeneration (2) was based on fabricated data (3). I recommend caution when taking these phenomena as absolute truths in neurodegeneration.

Response: Thank you for pointing this out to us. We have revised the noted section as follows and lowered the confidence level.

“Therefore, tau protein accumulation in MPS suggests impaired autophagy [92], which can also negatively impact neurological symptoms, as shown in several other diseases, such as Alzheimer, Parkinson, and Huntington diseases [93–95]. β-amyloid accumulation would also have a negative effect on the brain, as shown in previous studies [96].”

4)     Lines 428-438. These paragraphs should be included in the legend of figure 1.

Response: Thank you for your suggestion on the figure legends. We have revised it as follows. To make it easier to distinguish the figure legends from the main text, we added a new line between them and changed the font size. In addition, to acknowledge the website used to create this figure, we have added the statement "This figure was created with Biorender.com." to the end of this figure legend.

5)     Lines 480-488. These paragraphs should be included in the legend of figure 2.

Response: To make it easier to distinguish the figure legends from the main text, we added a new line between them and changed the font size.

6)     Lines 539-544. These paragraphs should be included in the legend of figure 3.

Response: To make it easier to distinguish the figure legends from the main text, we added a new line between them and changed the font size.

7)     Lines 586-587. The authors employ the plural although just one Clinical Trial is cited. As the authors state in section 4.2, only one clinical trial is discussed in each subsection 4.2.x, and that is perfectly reasonable. However, in these lines, it would be interesting to cite other ongoing trials, if any. Besides that, it would be beneficial to include more details about the clinical trial discussed by the authors, specially concerning the present phase of the trial and main results thus far.

Response: I appreciate your suggestion. We have added explanations to the pointed part according to your instructions.

“Currently, MPS IIIA is the only type employing scAAV in clinical trials (ClinicalTrials.gov Identifier: NCT04088734, phase 1/2, terminated due to lack of efficacy in patients with cognitive Development Quotient (DQ) below 60; NCT02716246, phase 2/3, ongoing in patients with DQ 60 or above), although it would not be impossible for other types to use scAAV vectors, judging from the size of each responsible gene.”

8)     Lines 642-649. These paragraphs should be included in the legend of figure 4.

Response: To make it easier to distinguish the figure legends from the main text, we added a new line between them and changed the font size.

9)     Lines 681-682. Are these Clinical Trials parts of the same trial? Again, it would be beneficial to include information about the trials’ phase in the text.

Response: Thank you for your question and suggestion. These two are different, not sequential. We have added the information about each clinical trial phase to the pointed part with a specific NCT number.

Additional refs

  1. Gurinova, E.E.; Maksimova, N.R.; Sukhomyasova, A.L. Clinical Description of a Rare Autosomal Recessive Syndrome in the Yakut Children. Yakut Med. J. 2014, 2, 14–18.
  2. Kondo H, Maksimova N, Otomo T, Kato H, Imai A, Asano Y, Kobayashi K, Nojima S, Nakaya A, Hamada Y, Irahara K, Gurinova E, Sukhomyasova A, Nogovicina A, Savvina M, Yoshimori T, Ozono K, Sakai N. Mutation in VPS33A affects metabolism of glycosaminoglycans: a new type of mucopolysaccharidosis with severe systemic symptoms. Hum Mol Genet. 2017 Jan 1;26(1):173-183. doi: 10.1093/hmg/ddw377.

  3. Dursun A, Yalnizoglu D, Gerdan OF, Yucel-Yilmaz D, Sagiroglu MS, Yuksel B, Gucer S, Sivri S, Ozgul RK. A probable new syndrome with the storage disease phenotype caused by the VPS33A gene mutation. Clin Dysmorphol. 2017 Jan;26(1):1-12. doi: 10.1097/MCD.0000000000000149.PMID: 27547915
  4. Vasilev F, Sukhomyasova A, Otomo T. Mucopolysaccharidosis-Plus Syndrome. Int J Mol Sci. 2020 Jan 9;21(2):421. doi: 10.3390/ijms21020421.PMID: 31936524